# EXPLICIT PARETO FRONT OPTIMIZATION
# FOR CONSTRAINED REINFORCEMENT LEARNING

## ABSTRACT

Many real-world problems require that reinforcement learning (RL) agents learn policies that not only maximize a scalar reward, but do so while meeting constraints, such as remaining below an energy consumption threshold. Typical approaches for solving constrained RL problems rely on Lagrangian relaxation, but these suffer from several limitations. We draw a connection between multi-objective RL and constrained RL, based on the key insight that the constraint-satisfying optimal policy must be Pareto optimal. This leads to a novel, multi-objective perspective for constrained RL. We propose a framework that uses a multi-objective RL algorithm to find a Pareto front of policies that trades off between the reward and constraint(s), and simultaneously searches along this front for constraint-satisfying policies. We show that in practice, an instantiation of our framework outperforms existing approaches on several challenging continuous control domains, both in terms of solution quality and sample efficiency, and enables flexibility in recovering a portion of the Pareto front rather than a single constraint-satisfying policy.

## 1 INTRODUCTION

Deep reinforcement learning (RL) has shown great potential for training policies that optimize a single scalar reward. Recent approaches have exceeded human-level performance on Atari (Mnih et al., 2015) and Go (Silver et al., 2016), and have also achieved impressive results in continuous control tasks, including robot locomotion (Lillicrap et al., 2016; Schulman et al., 2017), acrobatics (Peng et al., 2018), and real-world robot manipulation (Levine et al., 2016; Zeng et al., 2019).

However, many problems, especially in the real world, require that policies meet certain constraints. For instance, we might want a factory robot to optimize task throughput while keeping actuator forces below a threshold, to limit wear-and-tear. Or, we might want to minimize energy usage for cooling a data center while ensuring that temperatures remain below some level (Lazic et al., 2018).

Such problems are often encoded as constrained Markov Decision Processes (CMDPs) (Altman, 1999), where the goal is to maximize task return while meeting the constraint(s). Typical approaches for solving CMDPs use Lagrangian relaxation (Bertsekas, 1999) to transform the constrained optimization problem into an unconstrained one. However, existing Lagrangian-based approaches suffer from several limitations. First, because the relaxed objective is a weighted-sum of the task return and constraint violation, this assumes a convex Pareto front (Das & Dennis, 1997). In addition, when the constraint is difficult to satisfy, in practice policies can struggle to obtain task reward. Finally, such approaches typically produce a single policy that satisfies a specific constraint threshold. However, the exact constraint threshold may not be known in advance, or one may prefer to choose from a set of policies across a range of acceptable thresholds.

We aim to achieve the goal of CMDPs (i.e., finding a constraint-satisfying policy that maximizes task return) while avoiding these limitations, by introducing a novel, general framework based on multi-objective MDPs (MO-MDP). A MO-MDP can be seen as a CMDP where the constrained objectives are instead unconstrained. Our key insight is that if we have access to the Pareto front, then we can find the optimal constraint-satisfying policy by simply searching along this front. However, finding the entire Pareto front is unnecessary if only a relatively small portion of the policies along the Pareto front meet the constraints. Therefore, we propose to also simultaneously prioritize learning for the preferences (i.e., trade-offs between reward and cost) that are most likely to produce policies that satisfy the constraints, and thus cover the relevant part of the Pareto front.

To our knowledge, there is no existing framework for applying multi-objective RL algorithms to constrained RL problems. Our main contribution is a general framework that enables this, by learning which preferences produce constraint-satisfying policies. This framework can be combined with any multi-objective RL algorithm that learns an approximate Pareto front of policies. Our second contribution is to extend a state-of-the-art multi-objective RL algorithm, multi-objective maximum a posteriori policy optimization (MO-MPO) (Abdolmaleki et al., 2020), to learn the Pareto front in a single training run. We use this extension of MO-MPO within our framework, and call the resulting algorithm **constrained MO-MPO**. We show in practice, constrained MO-MPO outperforms existing approaches on challenging continuous control tasks with constraints.

## 2 RELATED WORK

**Constrained reinforcement learning.** Constrained RL algorithms seek policies that meet the desired constraints at deployment time. Most approaches use Lagrangian relaxation (Bertsekas, 1999). Recent Lagrangian-based approaches claim convergence to constraint-satisfying (Tessler et al., 2019) or optimal (Paternain et al., 2019) solutions, but this is debatable (Szepesvari, 2020). Recent works seek to stabilize this optimization by approximating the reward and cost functions with convex relaxations (Yu et al., 2019) or by utilizing derivatives of the constraint function (Stooke et al., 2020). Other works have applied Lagrangian relaxation to mean-value constraints (Tessler et al., 2019), convex set constraints (Miryoosefi et al., 2019), and local constraint satisfaction (Bohez et al., 2019).

Existing Lagrangian approaches involve linear scalarization, however, and thus cannot find solutions that lie on concave portions of the true Pareto front (Das & Dennis, 1997). In contrast, we build on a multi-objective RL algorithm that does not rely on scalarization, and in practice, our approach can indeed find constraint-satisfying solutions on a concave Pareto front (see Sec. 5.1, humanoid **walk**).

**Safe reinforcement learning.** In safe RL, the aim is to achieve constraint satisfaction not only during deployment, but also during learning (García & Fernández, 2015). Recent works for deep RL modify the policy improvement step to guarantee that the policy will never violate constraints during training (Achiam et al., 2017; Berkenkamp et al., 2017; Chow et al., 2018; 2019; Yang et al., 2020; Zhang et al., 2020). These approaches require, however, that the initial policy meets (or almost meets) the constraints. Otherwise, performance degrades substantially and constrained policy optimization (CPO) (Achiam et al., 2017), for example, performs worse than Lagrangian-based approaches (Ray et al., 2019). The aim of our work is to find better solutions for constrained RL, rather than safe RL. We discuss in Sec. 6 how our method can be extended to reduce constraint violation during training.

**Multi-objective reinforcement learning.** Our approach is built on ideas from multi-objective RL (MORL), which consists of single-policy and multi-policy approaches. Single policy methods learn a policy that is optimal for a given setting of reward preferences. Most rely on linear scalarization (Roijers et al., 2013), which restricts solutions to the convex portions of the Pareto front and can be sensitive to reward scales. Non-linear scalarizations have been proposed (Tesauro et al., 2008; Van Moffaert et al., 2013; Golovin & Zhang, 2020), but these are harder to combine with value-based RL and have seen limited use in deep RL. Recently, Abdolmaleki et al. (2020) introduced MO-MPO, where the preferences represent per-objective constraints on the policy improvement step. MO-MPO does not rely on scalarization and is thus invariant to reward scales.

Multi-policy MORL aims to find a *set* of policies that covers the whole Pareto front. Recent approaches learn a manifold in parameter space that optimizes the hypervolume of the Pareto front (Parisi et al., 2016; 2017). While such approaches could be combined with our framework, their scalability to deep RL remains to be shown. Other works combine single policy approaches with a general objective to optimize hypervolume (Xu et al., 2020). The instantiation of our framework is similar in spirit to such two-level methods, but applied to a different problem setting.

## 3 BACKGROUND AND NOTATION

### 3.1 CONSTRAINED MARKOV DECISION PROCESSES

A constrained Markov Decision Process (CMDP) consists of states $s \in \mathcal{S}$, actions $a \in \mathcal{A}$, an initial state distribution $p_0(s)$, a transition function $p(s'|s, a)$, reward functions $\{r_k(s, a)\}_{k=0}^K$, constraint

thresholds $\{c_k\}_{k=1}^K$, and a discount factor $\gamma \in [0, 1]$. The $(K + 1)$ reward functions consist of a task reward $r_0$ and constrained rewards $r_{1:K}$; we will refer to these together as *objectives*.

A policy $\pi(a|s)$ maps from a given state to a distribution over actions. The optimal solution to a constrained MDP is a policy that maximizes the expected return for the task reward $r_0$, while ensuring that the expected return for all constrained rewards $r_{1:K}$ satisfy their respective thresholds:

$$\max_\pi \ \mathbb{E}_\pi\Big[\sum_t \gamma^t r_0(s_t, a_t)\Big] \quad \text{s.t.} \ \mathbb{E}_\pi\Big[\sum_t \gamma^t r_i(s_t, a_t)\Big] \geq c_i \quad \forall i = 1, ..., K \,, \tag{1}$$

where $\mathbb{E}_\pi$ is shorthand for the expectation over trajectories when following the policy $\pi$, given a fixed initial state distribution. Here the constraint thresholds are lower bounds, without loss of generality.

The value function $V_k^\pi(s)$ of a policy $\pi$ is its expected return for objective $r_k$ when starting from state $s$. The action-value function $Q_k^\pi(s, a)$ denotes the expected return for objective $r_k$ after taking action $a$ in state $s$ and thereafter acting according to the policy: $Q_k^\pi(s, a) = r_k(s, a) + \gamma \mathbb{E}_{s' \sim p(s'|s,a)}[V_k^\pi(s')]$.

Most approaches for solving CMDPs are based on Lagrangian relaxation, where the constrained problem is turned into the following unconstrained optimization problem:

$$\min_{\lambda \geq 0} \max_\pi \ \mathbb{E}_\pi\Big[\sum_t \gamma^t r_0(s_t, a_t)\Big] + \sum_{k=1}^K \lambda_k \Big(\mathbb{E}_\pi\Big[\sum_t \gamma^t r_k(s_t, a_t)\Big] - c_i\Big). \tag{2}$$

These approaches alternate between optimizing for the Lagrange multipliers $\lambda_k$ and the policy $\pi$.

While our multi-objective perspective on constrained RL aims to achieve the goal of CMDPs—finding a constraint-satisfying policy that maximizes task return—it does not directly solve (1). Nonetheless, we will show our approach outperforms those that directly solve (1) via Lagrangian relaxation.

## 3.2 MULTI-OBJECTIVE REINFORCEMENT LEARNING

In order to achieve the goal of CMDPs, we propose to leverage advances in multi-objective RL. A multi-objective MDP (MO-MDP) is defined in the same way as a CMDP, except without constraint thresholds. Because there are multiple unconstrained objectives, there is not a single optimal policy. Instead, there is a set of optimal policies, called the *Pareto front*. A policy is *Pareto optimal* if there is no other policy that improves its return for one objective without decreasing return for another.

**Preferences.** Each policy on the Pareto front is the optimal policy for a particular setting of preferences (i.e., desired trade-off over objectives).[1] This is typically encoded via a preference vector $\epsilon$, in which each element $\epsilon_k$ represents the relative importance of the corresponding objective $r_k$.

**Connection to CMDPs.** Given a CMDP $\{\mathcal{S}, \mathcal{A}, p_0, p, r_{0:K}, c_{1:K}, \gamma\}$, consider the corresponding MO-MDP $\{\mathcal{S}, \mathcal{A}, p_0, p, r_{0:K}, \gamma\}$. Regardless of the thresholds $c_{1:K}$, the optimal policy for the CMDP must lie on the Pareto front for this corresponding MO-MDP.[2] Thus, if we knew the Pareto front, we could simply search along it to find the constraint-satisfying policy that maximizes task return.

**Multi-objective Maximum a Posteriori Policy Optimization (MO-MPO).** Whereas most MORL algorithms are based on linear scalarization, MO-MPO takes a *distributional* approach to training policies. For each policy improvement step, MO-MPO first computes a non-parametric policy *for each objective* $r_k$ that improves the (parametric) policy with respect to that objective, subject to a non-negative constraint $\epsilon_k$ on the KL-divergence between the improved and old policies. Then, the policy is updated via supervised learning on the sum of these non-parametric policies. Intuitively, $\epsilon_k$ defines the influence of objective $r_k$ on the final policy; an objective with larger $\epsilon_k$ has more influence. However, MO-MPO can only train policies for a single preference setting $\epsilon$. Sec. 4.1 explains how we extend MO-MPO to train preference-conditioned policies $\pi(a|s, \epsilon)$, in order to use it in our framework. We choose to extend MO-MPO because it does not suffer from the limitations of linear scalarization, and has been shown to outperform such approaches (Abdolmaleki et al., 2020).

---

[1]This assumes that the choice of preference encoding does not restrict optimal solutions to the convex portions of the Pareto front. Although linear scalarization does not satisfy this requirement, other kinds of preference encodings do, for instance Chebyshev scalarization (Van Moffaert et al., 2013) or the per-objective KL-divergence constraints in MO-MPO, described later in this section.

[2]This is because if the optimal policy for the CMDP is not Pareto optimal, then there must exist another policy that obtains higher task return while meeting the constraints, which leads to a contradiction.

# 4 APPROACH

**Problem Statement.** For a given CMDP, we seek to find the preference vectors $\epsilon$ in the corresponding MO-MDP that produce constraint-satisfying action policies $\pi(a|s, \epsilon)$. To do this, we want to find the distribution $\pi(\epsilon)$ that maximizes the probability that the action policies satisfy the constraints.[3]

**Overview of Framework.** Our framework enables applying multi-objective RL algorithms to solve constrained RL problems. It is inspired by the observation that for any CMDP, there exist preference settings $\epsilon$ for the objectives that lead to the desired constraint-satisfying solutions. We propose to find these preference settings by optimizing for a hierarchical policy $\pi_\psi(\epsilon)\pi_\theta(a|s, \epsilon)$ that first selects a preference setting $\epsilon$ and then takes an action conditioned on $\epsilon$. We also maintain a policy evaluation function $Q_k^\pi(a, s, \epsilon)$ for each objective.

To train this policy, we decouple learning the preference policy $\pi_\psi(\epsilon)$ from learning the action policy $\pi_\theta(a|s, \epsilon)$. Formally, we alternate between optimizing two sub-problems:

1. Learn a new action policy $\pi_{\text{new}}(a|s, \epsilon)$ given the current distribution of preferences $\pi_{\text{old}}(\epsilon)$.
2. Learn a new preference policy $\pi_{\text{new}}(\epsilon)$ given the current action policy $\pi_{\text{old}}(a|s, \epsilon)$ .

We aim to converge to an optimal policy $\pi^*(\epsilon)\pi^*(a|s, \epsilon)$ that not only satisfies the constraints but is also Pareto optimal. Note that we can use different time scales for alternating between Steps 1 and 2. One extreme is to first fully optimize for a Pareto front curve given an initial preference distribution, and then fully optimize for solutions that satisfy the constraints. In our experiments we take the other extreme: we alternate between taking one learning step on $\pi_\psi(\epsilon)$ and $\pi_\theta(a|s, \epsilon)$ independently.

For Step 1, any multi-objective RL algorithm capable of training preference-conditioned policies can be used. Sec. 4.1 describes how we use MO-MPO for Step 1, which requires non-trivially extending it to train preference-conditioned policies. For Step 2, the notion of learning a preference policy is the core novel component of our framework: this is what enables us to apply multi-objective RL algorithms to constrained RL problems. Sec. 4.2 describes how to do this.

## 4.1 LEARNING PREFERENCE-CONDITIONED ACTION POLICIES (STEP 1)

In this section, we explain how we extend MO-MPO (Abdolmaleki et al., 2020) to learn preference-conditioned policies for the current preference distribution $\pi_{\text{old}}(\epsilon)$.

MO-MPO can only train policies $\pi(a|s)$ for a single preference $\epsilon$, so it cannot be directly used. This section describes how we extend MO-MPO to train a *single* policy $\pi_\theta(a|s, \epsilon)$ conditioned on preference parameters $\epsilon \sim \pi_{\text{old}}(\epsilon)$, that represents the entire Pareto front. This is a non-trivial extension, that requires making both the action policy and Q-functions preference-conditioned, and replacing the scalar temperature with a preference-conditioned temperature network, as well as modifying the underlying MO-MPO optimization principles. We also introduce hindsight relabeling of preferences to stabilize off-policy learning and improve sample efficiency.

Our extended version is a policy iteration algorithm with two steps:

- Policy evaluation: learn preference-conditioned Q-functions for all objectives.
- Policy improvement: improve the preference-conditioned policy according to Q-functions.

**Preference-conditioned policy evaluation.** In this step, we train a separate Q-function per objective (coined Q-decomposition by Russell & Zimdars 2003), to evaluate the current policy $\pi_{\text{old}}(\epsilon)\pi_{\text{old}}(a|s, \epsilon)$ under the state distribution $\mu(s)$. To learn the Q-function, for each learning step we sample $L$ transitions from the replay buffer $\{s_i, a_i, \{r_i^k\}_k^K, s_i'\}_i^L$. Since our Q-functions $Q_k^\pi(a, s, \epsilon)$ are also conditioned on preferences, we use hindsight relabeling to augment the states with preference parameters sampled from the current preference policy $\epsilon_i \sim \pi_{\text{old}}(\epsilon)$, resulting in transitions $\{[s_i, \epsilon_i], a_i, \{r_i^k\}_k^K, [s_i', \epsilon_i]\}_i^L$. Any policy evaluation algorithm can be used to learn these Q-functions. We use distributional policy evaluation with 5-step return (Barth-Maron et al., 2018).

**Preference-conditioned policy improvement.** The policy improvement step assumes a state distribution $\mu(s)$, per-objective Q functions $Q_k^{\text{old}}(s, a, \epsilon)$, the current action policy $\pi_{\text{old}}(a|s, \epsilon)$, and the current preference policy $\pi_{\text{old}}(\epsilon)$. It consists of two sub-steps: 1) finding per-objective improved

---

[3]Without loss of generality, we consider a state-independent preference policy $\pi(\epsilon)$ when deriving the update rules. In Sec. 5.3, we show our approach can also be used to learn state-dependent preference policies $\pi(\epsilon|s)$.

action distributions, and 2) distilling these distributions into a new preference-conditioned action policy via supervised learning.

*Finding per-objective distributions:* To find per-objective improved action distributions $q_k(a|s, \epsilon)$, we optimize the following RL optimization problem for each objective:

$$\max_{q_k} \quad \mathbb{E}_{\pi_{\text{old}}(\epsilon)\mu(s)} \Big[ \int_a q_k(a|s, \epsilon) \, Q_k^{\text{old}}(s, a, \epsilon) \, da \Big] \tag{3}$$

$$\text{s.t.} \quad \mathbb{E}_{\mu(s)} \Big[ \text{KL}(q_k(a|s, \epsilon) \| \pi_{\text{old}}(a|s, \epsilon)) \Big] < \epsilon_k \quad \forall \, \epsilon \sim \pi_{\text{old}}(\epsilon) \,.$$

We can solve this problem in closed form to obtain

$$q_k(a|s, \epsilon) \propto \pi_{\text{old}}(a|s, \epsilon) \exp \Big( \frac{Q_k^{\text{old}}(s, a, \epsilon)}{\eta_{\omega_k}(\epsilon_k)} \Big), \tag{4}$$

where $\eta_{\omega_k}(\epsilon_k)$ is a preference-dependent temperature function for objective $r_k$, and is parameterized by $\omega_k$. This temperature function is obtained by minimizing the following dual function:

$$g(\omega_k) = \mathbb{E}_{\pi_{\text{old}}(\epsilon)\mu(s)} \Big[ \eta_{\omega_k}(\epsilon_k) \Big( \epsilon_k + \log \int_a \pi_{\text{old}}(a|s, \epsilon) \exp \Big( \frac{Q_k^{\text{old}}(s, a, \epsilon)}{\eta_{\omega_k}(\epsilon_k)} \Big) da \Big) \Big]. \tag{5}$$

In practice, we maintain a single function $\eta_\omega(\epsilon)$ with shared parameters $\omega$ for all objectives.[4]

To approximate the expectations over the preference distribution and state distribution, we draw $L$ states from the replay buffer and $L$ preferences $\epsilon$ from the preference policy $\pi_{\text{old}}(\epsilon)$. To approximate the integrals over $a$, for each $(s, \epsilon)$ pair we sample $M$ actions from the current policy $\pi_{\text{old}}(a|s, \epsilon)$.

*Learning a new parameterized action policy:* After obtaining per-objective improved policies, we use supervised learning to distill these distributions into a new parameterized policy:

$$\max_\theta \sum_{k=0}^K \mathbb{E}_{\pi_{\text{old}}(\epsilon)\mu(s)} \big[ \text{KL}(q_k(a|s, \epsilon) \| \pi_\theta(a|s, \epsilon)) \big] \text{ s.t. } \mathbb{E}_{\pi_{\text{old}}(\epsilon)\mu(s)} \big[ \text{KL}(\pi_{\text{old}}(a|s, \epsilon) \| \pi_\theta(a|s, \epsilon)) \big] < \beta,$$

subject to a trust region with bound $\beta > 0$ for more stable learning. To solve this optimization, we use Lagrangian relaxation as described in Abdolmaleki et al. (2018; 2020).

## 4.2 LEARNING PREFERENCE POLICIES (STEP 2)

We will now explain how to optimize the preference distribution $\pi_\psi(\epsilon)$ for selecting action policies $\pi_{\text{old}}(a|s, \epsilon)$ with better constraint satisfaction. To achieve this, we define a fitness function $f_k$ that evaluates satisfaction of constraint threshold $c_k$, given the current action policy $\pi_{\text{old}}(a|s, \epsilon)$ and Q-function $Q_k^{\text{old}}(s, a, \epsilon)$. The fitness function should be chosen based on the problem and form of the constraint. In our empirical evaluation, for equality constraints we use a fitness function that penalizes the difference between the expected Q-values and constraint threshold, and for inequality constraints it penalizes the amount the expected Q-values fall under the threshold:

$$f_k^{\text{eq}}(\epsilon) = -\Big| \mathbb{E}_{\mu(s)} \Big[ Q_k^{\text{old}}(s, \mathbb{E}_{\pi_{\text{old}}(a|s, \epsilon)}[a], \epsilon) \Big] - c_k \Big| \tag{6}$$

$$f_k^{\text{ineq}}(\epsilon) = \min \Big( 0, \, \mathbb{E}_{\mu(s)} \Big[ Q_k^{\text{old}}(s, \mathbb{E}_{\pi_{\text{old}}(a|s, \epsilon)}[a], \epsilon) \Big] - c_k \Big). \tag{7}$$

Given a chosen fitness function, we now can use any off-the-shelf RL algorithm to optimize the preference policy $\pi_\psi(\epsilon)$, initialized by $\pi_{\text{old}}(\epsilon)$, to maximize the fitness function. See Appendix D for more details on the general procedure underlying our algorithm (Appendix, Algorithm 1).

## 5 EXPERIMENTS

We find that our approach of Pareto front optimization applied to MO-MPO (i.e., constrained MO-MPO), finds solutions that are on par with those found by MORL algorithms (Appendix C.1). In

---

[4]The optimization problem in (3) generalizes the one in MO-MPO, which assumes fixed KL-constraints $\epsilon$, to a distribution $\pi_{\text{old}}(\epsilon)$ over KL-constraints. Thus (4) and (5) can be derived by following steps analogous to those given in Abdolmaleki et al. (2020).

our experiments, we compare constrained MO-MPO against existing RL algorithms for constrained MDPs: for a number of challenging continuous control tasks, we analyze the quality of policies found for a range of constraint thresholds. Unless otherwise mentioned, we use (6) as the fitness function. For action policies we use Gaussian distributions parameterized by neural networks, and for the preference policy in constrained MO-MPO, we use a discrete distribution. Architecture details and learning hyperparameters are described in Appendix A.

**Domains.** We evaluate our approach in two continuous control domains. First, we use the humanoid *run* and *walk* tasks from DeepMind Control Suite (Tassa et al., 2018). This domain has a challenging 21-dimensional action space. The agent receives a shaped reward for running at 10 m/s or walking at 1 m/s. The constraint is imposed on the expected negative control norm (i.e., $-\|a\|_2$), which roughly captures the "energy" expended by the agent. This is relevant for real-world settings (e.g., in robotics or control tasks), where energy consumption often needs to be taken into account.

We also evaluate on the level-two point mass tasks from the Safety Gym suite (Ray et al., 2019), which we refer to as point *goal*, *button*, and *push*. The agent receives a sparse reward for either reaching a goal location or pushing a box to a goal. The constraint is imposed on the expected per-episode cost, which is incurred by running into or over objects. There are several types of objects, each with its own cost. Unless otherwise mentioned, the constraint is with respect to the total cost.

**Pareto plots.** When we plot learned Pareto fronts for the tasks (e.g., in Fig. 1, top), the x-coordinate is average per-episode negative cumulative cost and the y-coordinate is average per-episode task reward.[5] For both axes, higher values are better. All trained policies for the same task are evaluated on the same set of randomly-initialized environments. For constrained MO-MPO policies, we sample a new preference at the beginning of each episode, rather than at every timestep.

**Baselines.** As a baseline we consider Lagrangian relaxation; because of its simplicity and effectiveness it is a common choice (Ray et al., 2019; Stooke et al., 2020). To ensure a fair comparison with our approach, we use MPO to train policies to optimize the Lagrangian dual objective (2). We obtain state-of-the-art results on Safety Gym tasks with this *MPO-Lagrangian* baseline (Appendix C.2).

## 5.1 Quality of solutions

We evaluated our approach and the baseline for constraint thresholds linearly spaced in the range $[-4, -1]$ for humanoid *run*, $[-2, -0.5]$ for humanoid *walk*, and $[-15, -2]$ for point tasks. For constrained MO-MPO, the preference policy $\pi(\epsilon)$ is initialized to uniform and is a discrete distribution over 100 linearly-spaced values starting at $10^{-5}$ and up $0.15$ for humanoid *run*, $0.3$ for humanoid *walk*, and $0.2$ for the point tasks. Results are shown in Fig. 1.

We observe the biggest difference in performance for humanoid *walk*: the Lagrangian baseline only finds policies at the extremes, whereas our approach finds constraint-satisfying solutions that achieve non-zero task reward. This may be because the ground-truth Pareto front for this task is concave (Sec. 2). On humanoid *run*, point *goal*, and point *button*, the baseline and our approach find similar-quality solutions for easier constraint thresholds, but as the constraint threshold becomes more difficult to satisfy (i.e., smaller in magnitude), our approach dominates (Fig. 1, top row; Fig. 2a). The only exception is point *push*, for which both approaches find similar-quality solutions for all constraint thresholds. We hypothesize that this is because in this particular task, the cost does not conflict much with the task objective, so it is easier to find high-task-reward solutions that incur small cost.

Our approach always finds policies that satisfy the constraints for humanoid, whereas the Lagrangian baseline occasionally violates the constraint.[6] For Safety Gym tasks, the policies found by both our approach and the baseline slightly violate the constraints (Fig. 1, bottom row); this is due to Q-function underestimates for the cost, rather than a limitation of the policy optimization (Appendix C.4).

---

[5] The action norm constraint for the humanoid tasks is per-timestep with 1000 timesteps per episode, so a policy that exactly meets a constraint threshold of $-2$, has an average cumulative episodic cost of $-2000$.

[6] With longer training, these violations might disappear, but we do not expect the task reward to improve. These violations are due to the relatively unstable behavior of the Lagrangian baseline: when the constraint starts to be violated, this leads to a gradual increase in the Lagrange multiplier, and it takes some time for this to propagate into changes in the policy. This can be seen empirically in Fig. 2b (top row) where the average cost dips below (i.e., violates) the constraint threshold, indicated by the dotted line.

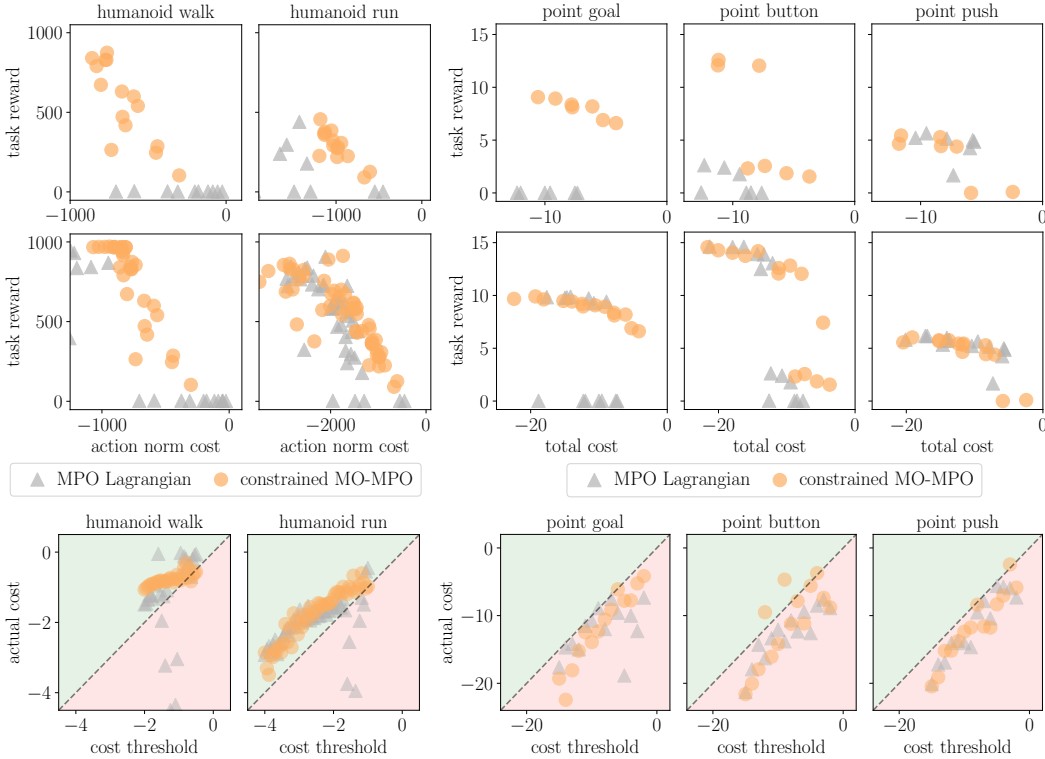

Figure 1: Top row: For harder-to-satisfy (i.e., lower magnitude) constraint thresholds, constrained MO-MPO finds policies with higher task performance than those found by the Lagrangian baseline, in all tasks except point **push**. Middle row: Across all constraint thresholds, constrained MO-MPO performs at least as well as the baseline. The plots in the top and middle row are Pareto plots: for both axes, higher values are better. Each dot corresponds to a separate policy trained for a particular constraint threshold. Bottom row: Constrained MO-MPO and MPO-Lagrangian perform comparably in learning policies that satisfy the constraints. Policies that lie in the red region (below the dotted line) violate the constraint.

Our approach is also significantly more sample-efficient than the baseline. For instance on humanoid **run**, constrained MO-MPO learns constraint-satisfying policies that reach reasonably high task reward mid-way through training, whereas the majority of MPO-Lagrangian policies obtain zero task reward at the same point (Fig. 3a, 3b). This could be because with two conflicting objectives, MPO-Lagrangian cannot optimize for both objectives at once, whereas our approach can (Fig. 2b).

**Multiple random seeds.** We additionally ran experiments with five random policy initializations per constraint threshold, for a subset of four thresholds. The results support the conclusions drawn above. In particular, our approach significantly outperforms the baseline for the harder-to-satisfy constraint thresholds, in four out of the five tasks (see Appendix C.3 for details).

**Scaling to multiple constraints.** We also investigated how well constrained MO-MPO scales to multiple constraints, for a more difficult version of point **goal** (see Appendix B.2 for details). There are two constraints: one on the cost of crossing over hazards, and one on bumping into vases. For each type of cost, we fix its threshold at $-2$ and vary the other between $-10$ and $0$. The baseline fails to obtain task reward when the constraint for hazards is fixed at the challenging threshold of $-2$, whereas constrained MO-MPO finds a range of solutions (Fig. 3c, right).

## 5.2 FLEXIBILITY

Since our approach decouples Pareto front optimization from Pareto exploration, by choosing the appropriate fitness function, one can in theory recover any portion of the Pareto front. This is useful because in practice, an RL practitioner may not know the true constraint, and will be better able to

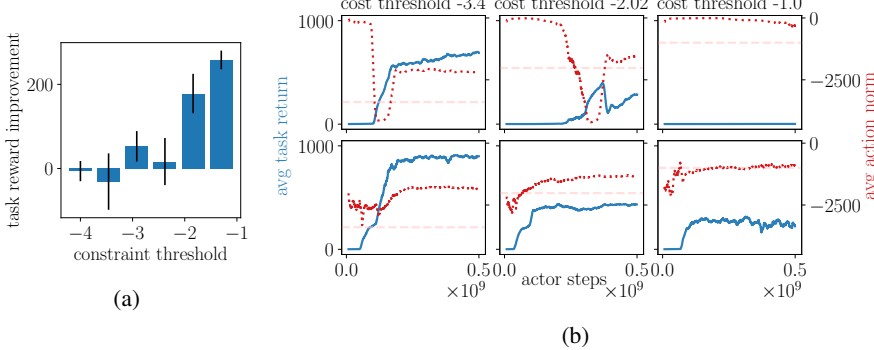

(a)

(b)

Figure 2: Plots are for humanoid *run*. (a) The difference in average reward obtained by constrained MO-MPO versus MPO-Lagrangian (positive values mean constrained MO-MPO performs better), across sub-ranges of constraint thresholds and with standard error bars. Our approach significantly outperforms the baseline for more difficult (lower magnitude) constraint thresholds. (b) Constrained MO-MPO (bottom) is able to optimize for both objectives at once, whereas the baseline (top) alternates between the two, leading to large drops in action norm cost. The horizontal dashed lines indicate the cost threshold.

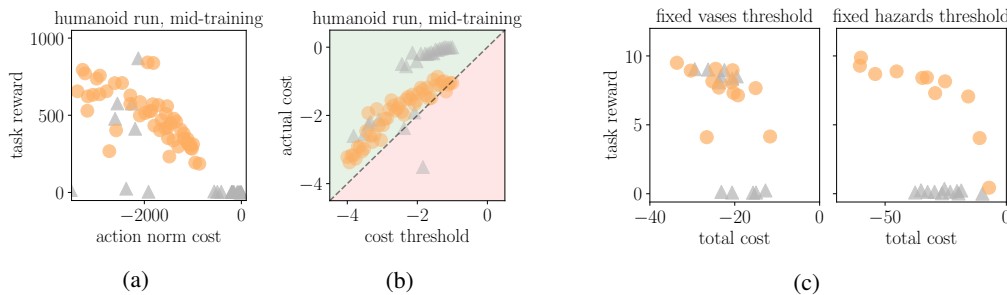

(a)          (b)          (c)

Figure 3: (a) Pareto fronts and (b) constraint violations for humanoid *run* from the middle of training. (c) Pareto fronts for point *goal* with two constraints. Refer to Fig. 1 for interpretation of colors.

choose which policy to deploy after observing a couple of options along the Pareto front. Typical approaches to constrained RL find a *single* policy that meets the constraint exactly.[7]

On humanoid *run*, we evaluate constrained MO-MPO using (7) as the fitness function for several inequality constraint thresholds. We find that our approach indeed learns a portion of the Pareto front that satisfies the constraint (Fig. 4a). These plots are obtained by taking the preferences at regular percentiles between 5th and 95th percentiles of the learned preference distribution, and evaluating the policy conditioned on each of these preferences, on the same set of randomly-initialized environments.

### 5.3 STATE-DEPENDENT PREFERENCES

By using a state-dependent preference policy $\pi(\epsilon|s)$, we can extend our approach to satisfy constraints *per-initial-state*, rather than in expectation.[8] To do this, we train a neural network that maps the state to a discrete distribution over preference settings. We implement this for the MPO-Lagrangian baseline by making the Lagrange multiplier state-dependent as well, as introduced by Bohez et al. (2019); this trains policies that satisfy the constraint *per-state*—there is no equivalent for the Lagrangian approach that satisfies constraints per-initial-state.

We use both approaches to train policies for humanoid *run*, for the same settings of constraint thresholds as in Sec. 5.1. Our approach finds constraint-satisfying policies that achieve higher task reward than the baseline (Fig. 4b). The constraint violation is computed by summing the per-episode

---

[7]Although it is possible to extend the Lagrangian baseline by conditioning the policy on the threshold, given that constrained MO-MPO finds better solutions, we choose to focus evaluation here on constrained MO-MPO.

[8]In our implementation, we sample a preference from the preference policy only at the start of every episode, rather than per-timestep. If we did the latter, then we could satisfy *per-state* constraints.

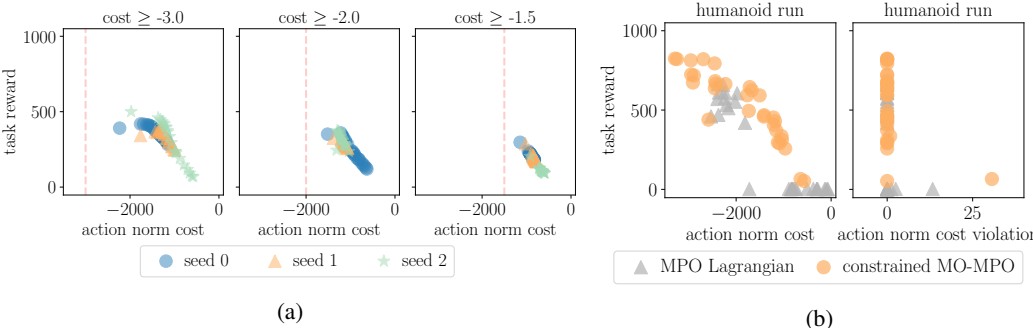

(a)                                              (b)

Figure 4: (a) Pareto plots for constrained MO-MPO on humanoid *run* for three inequality constraints, with three random initializations for each. The vertical dashed lines indicate the cost threshold. (b) Policies trained by constrained MO-MPO with *per-initial-state* constraint satisfaction obtain higher task reward, while satisfying the constraints.

constraint violation, obtained by comparing the average per-timestep action norm cost for that episode against the constraint threshold.

## 6   CONCLUSION AND FUTURE WORK

In this paper, we introduced a novel framework for constrained RL, by leveraging the ability of multi-objective RL algorithms to find Pareto-optimal solutions. This framework can be combined with any algorithm that finds a Pareto front of solutions. Our empirical results show that an instantiation of this framework, constrained MO-MPO, outperforms the commonly-used Lagrangian relaxation approach in terms of solution quality, stability, and sample-efficiency. In particular, the Lagrangian approach struggles to train policies that obtain task reward when the constraints are difficult to meet, or when the ground-truth Pareto front is concave. Our framework also enables flexibility via the choice of fitness function and can be extended to meet constraints per-initial-state, rather than in expectation.

One limitation of this work is that the support of the preference policy's discrete distribution must include preference setting(s) that satisfy the constraint threshold. In practice, this is straightforward to do for MO-MPO, because the $\epsilon_k$ encode preferences in a way that is independent of the objectives' reward scales. In future work, it is worth exploring other distributions for the preference policy that would overcome this limitation, for instance a mixture of Gaussians.

Another limitation is this work cannot be applied directly to safe RL. We plan to extend our approach to reduce the cost incurred during training, perhaps by initializing with a conservative distribution over preferences, that prioritizes minimizing cost. Finally, we also plan to exploit the flexibility of our framework to solve constrained RL problems with more than one unconstrained objective.

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

## A  IMPLEMENTATION DETAILS

We use an asynchronous off-policy actor-learner setup. In this setup, actors fetch policy parameters from the learner and act in the environment, writing transitions to the replay buffer. The learner uses the transitions in the replay buffer to update the policies and Q-functions (and optionally the temperature network and Lagrange multiplier network, depending on the approach). We implement this setup in Acme (Hoffman et al., 2020), a framework for distributed RL. To make learning more stable, we maintain a target network for each trained network; we use these target networks for computing gradients, and update them after every 200 gradient steps. We use 32 actors for training. We use Adam (Kingma & Ba, 2015) for optimization, with a learning rate of $10^{-4}$ unless otherwise specified.

All policies and Q-functions are implemented as feed-forward neural networks. Our action policies $\pi_\theta$ output a Gaussian distribution with a diagonal covariance matrix, and our preference policies $\pi_\psi$ output a categorical distribution. We found that adding layer normalization followed by a hyperbolic tangent ($\tanh$) to the first layer of the policy and Q-function networks improves stability of learning. For the Safety Gym point mass tasks, we train a separate Q-function per objective, with a shared first hidden layer. For humanoid tasks, the Q-values for action norm cost are instead computed exactly from the actions; thus these Q-values have a discount factor of zero.

The default hyperparameters we used for our experiments are reported in Table 1, and task-specific hyperparameters are reported in Table 2. In our experiments, we trained policies with constrained MO-MPO and the MPO Lagrangian baseline for a range of different constraint thresholds, these thresholds are reported in Table 3. We consider the "harder" constraint thresholds (as plotted in Fig. 1, top row) to be the 15 lowest-magnitude thresholds for the humanoid tasks, and the 7 lowest-magnitude thresholds for the point mass tasks.

**Gathering data.**  We gather data for the replay buffer via the actors. When the action policy $\pi_\theta$ is preference-conditioned, at the start of each episode, each actor first samples a preference $\epsilon'$ from the preference policy $\pi_\psi$, and then acts according to $\pi_\theta(a|s, \epsilon')$ until the end of the episode. For the next episode, the actor repeats this, sampling a new preference.

**Evaluation.**  For humanoid tasks, we evaluate each trained policy on the same 100 randomly-initialized environments. For Safety Gym tasks, we do the same on the same 500 randomly-initialized environments. Table 4 reports how many actor steps policies are trained for, before evaluation. We evaluate with a deterministic action policy $\pi_\theta$ by using the mean of the Gaussian distribution over actions. When the action policy is conditioned on preferences, at the beginning of each episode during evaluation, we first sample a preference $\epsilon$ from the preference policy, and then execute the mean of $\pi_\theta(a|s, \epsilon)$, sticking with the same sampled preference for the entire episode. However, when the preference policy is state-dependent, we instead condition the action policy on the median preference from $\pi_\psi(\epsilon|s)$, because we find that this leads to better satisfaction of per-episode constraints.

## B  EXPERIMENTAL DOMAINS

### B.1  HUMANOID

We use the humanoid ***run*** and ***walk*** tasks from DeepMind Control Suite (Tassa et al., 2018).[9] The observation space is 67-dimensional and the action space is 21-dimensional. Observations consist of joint angles, joint velocities, center-of-mass velocity, head height, torso orientation, and hand and feet positions. Actions correspond to joint accelerations; the minimum and maximum action limits are $-1$ and $1$, respectively. Each episode is 1000 timesteps.

The unconstrained task reward is given by the environment: there is a shaped reward for maintaining a horizontal speed (in any direction) of 10 m/s for humanoid ***run*** and 1 m/s for humanoid ***walk***. The shaped reward is equal to $\min(h/h^*, 1)$, where $h$ is the speed of the agent and $h^*$ is the target speed, both in m/s. The constrained reward is the negative $l2$-norm of the action vector: $r_{\text{cost}}(s, a) = -\|a\|_2$. This can be thought of as limiting the energy usage of the agent.

---

[9]Available at github.com/deepmind/dm_control.

| Hyperparameter | Default |
|---|---|
| Q-function network(s): $Q_k(s, a)$ or $Q_k(s, a, \epsilon)$ | |
|     layer sizes | $(512, 512, 256)$ |
|     support | $[-150, 150]$ |
|     number of atoms | 51 |
|     n-step returns | 5 |
|     discount factor $\gamma$ | 0.99 |
| action policy network: $\pi_\theta(a|s)$ or $\pi_\theta(a|s, \epsilon)$ | |
|     layer sizes | $(256, 256, 256)$ |
|     minimum variance | $10^{-12}$ |
|     maximum variance | unbounded |
| preference policy network: $\pi_\psi(\epsilon)$ or $\pi_\psi(\epsilon|s)$ | |
|     layer sizes for $\pi_\psi(\epsilon|s)$ | $(256, 256, 256)$ |
|     layer sizes for temperature network $\eta_\omega(\epsilon)$ | $(256, 256, 256)$ |
|     support for $\epsilon_{\text{task}}$ | $[0.1, 0.1]$ |
|     support for $\epsilon_{\text{cost}}$ | $[10^{-5}, 0.15]$ |
|     number of atoms | 100 |
| both policy networks and Q-function networks | |
|     layer norm on first layer? | yes |
|     $\tanh$ on output of layer norm? | yes |
|     activation (after each hidden layer) | ELU |
| Lagrange multipliers: $\lambda$ or $g(\lambda|s)$ | |
|     layer sizes for $g(\lambda|s)$ | $(256, 256, 256)$ |
|     softmax on output? | yes |
|     initial Lagrange multiplier $\lambda$ (before softmax) | 0 |
|     Adam learning rate | $10^{-5}$ |
| MPO / MO-MPO for action policy network $\pi_\theta$ | |
|     actions sampled per state | 20 |
|     default $\epsilon_k$ | 0.1 |
|     KL-constraint on policy mean, $\beta_\mu$ | $10^{-3}$ |
|     KL-constraint on policy covariance, $\beta_\Sigma$ | $10^{-7}$ |
|     initial temperature $\eta$ | 5 |
|     Adam learning rate (for dual variables) | $10^{-2}$ |
| MPO / MO-MPO for preference policy network $\pi_\psi$ | |
|     actions sampled per state | 20 |
|     default $\epsilon_k$ | 0.1 |
|     KL-constraint on policy, $\beta$ | $10^{-7}$ |
|     initial temperature $\eta$ | 5 |
|     Adam learning rate (for dual variables) | $10^{-2}$ |
| training | |
|     batch size | 512 |
|     replay buffer size | $10^6$ |
|     target network update period | 200 |

Table 1: Default hyperparameters for all approaches, with decoupled update on mean and covariance of the action policy.

## B.2 SAFETY GYM

We use the level-two point mass tasks in Safety Gym: *goal*, *button*, and *push* (Ray et al., 2019).[10] The action space is 2-dimensional, with minimum and maximum action limits of $-1$ and 1, respectively. The observation space is 60-dimensional for point *goal*, and 76-dimensional for point *button* and *push*. There are 1000 timesteps per episode. At the start of each episode, the entities (i.e., goal, agent, obstacles) are randomly initialized.

---

[10]Available at github.com/openai/safety-gym.

**Humanoid *run***

| | |
|---|---|
| MPO / MO-MPO for preference policy network $\pi_\psi$ | |
| KL-constraint on policy, $\beta$ | $10^{-8}$ |

**Humanoid *walk***

| | |
|---|---|
| preference policy network: $\pi_\psi(\boldsymbol{\epsilon})$ or $\pi_\psi(\boldsymbol{\epsilon}\|s)$ | |
| support for $\epsilon_{\text{cost}}$ | $[10^{-5}, 0.30]$ |
| MPO / MO-MPO for preference policy network $\pi_\psi$ | |
| KL-constraint on policy, $\beta$ | $10^{-6}$ |

**Safety Gym point *goal*, *button*, *push***

| | |
|---|---|
| preference policy network: $\pi_\psi(\boldsymbol{\epsilon})$ or $\pi_\psi(\boldsymbol{\epsilon}\|s)$ | |
| support for $\epsilon_{\text{cost}}$ | $[10^{-5}, 0.20]$ |

Table 2: Hyperparameters for humanoid and Safety Gym experiments that differ from the defaults in Table 1.

| Task | Constraint thresholds |
|---|---|
| Humanoid ***run*** | $c_{\text{cost}} \in \text{linspace}(-4.0, -1.0, 31)$ |
| Humanoid ***walk*** | $c_{\text{cost}} \in \text{linspace}(-2.0, -0.5, 31)$ |
| Safety Gym point ***goal***, ***button***, ***push*** | $c_{\text{cost}} \in \text{linspace}(-15, -2, 14)$ |
| Safety Gym point ***goal*** with two constraints | $c_{\text{hazard}} = -2, c_{\text{vase}} \in \text{linspace}(-10, 0, 11)$ |
| | $c_{\text{vase}} = -2, c_{\text{hazard}} \in \text{linspace}(-10, 0, 11)$ |

Table 3: The constraint thresholds that policies are trained on for each task.

We use the default task configurations as used in Ray et al. (2019), for instance how many obstacles are spawned, the regions in which entities can be spawned, the size of entities, and the type of observations. The only difference is that we use sparse task reward instead of shaped task reward, since the former is more realistic.

A sparse task reward of 1 is given in ***goal*** when the agent enters the goal area, in ***button*** when the agent touches the target button, and in ***push*** when the agent pushes a box into the goal area. The constrained reward is the negative cumulative cost. In ***goal*** the agent incurs cost by being in hazardous regions or bumping into vases; in ***button*** the agent incurs cost by being in hazardous regions, bumping into gremlins, or touching non-target buttons; in ***push*** the agent incurs cost being in hazardous regions or bumping into pillars. All obstacles are static, except for gremlins, which move in a fixed circular pattern; the agent moves vases when it bumps into them.

In our experiments with two constraints (Sec. 5.1), we use a more difficult variant of the point ***goal*** task, with two changes. First, the length of the square area in which entities are spawned is three-quarters of the original length, which makes the density of obstacles higher and thus more difficult for the agent to navigate around. In addition, the radius of the goal region is two-thirds the original radius, which allows for it to be placed in more difficult-to-reach locations: because the random initialization of the scene enforces that there are no collisions between the entities, the smaller the goal is, the more potential open areas it could be placed in.

## C   ADDITIONAL EXPERIMENTS AND ANALYSIS

### C.1   COMPARISON TO MULTI-OBJECTIVE RL ALGORITHMS

We first evaluate whether constrained MO-MPO is able to find solutions that lie on the Pareto front. Since the ground-truth Pareto front is not available in general, we approximate this for humanoid ***run***

| Task | Number of actor steps |
|------|----------------------|
| Humanoid *run* and *walk* | 500 million |
| Humanoid *run*, part-way through training | 200 million |
| Humanoid *run*, per-episode constraint satisfaction | 1 billion |
| Safety Gym point *goal*, *button*, *push* | 400 million |
| Safety Gym point *goal* with two constraints | 200 million |

Table 4: The number of actor steps that policies are trained for, before evaluation.

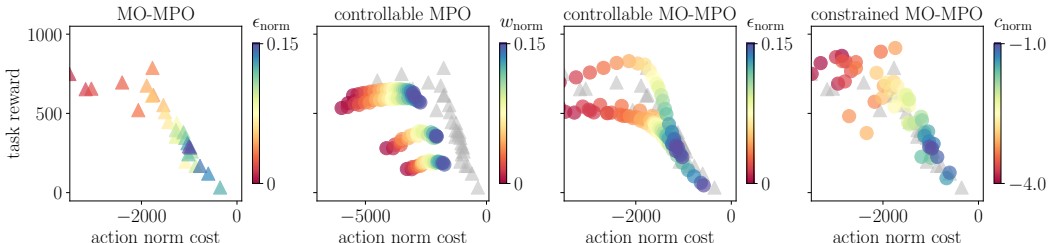

Figure 5: A comparison of the Pareto fronts for humanoid *run* found by MO-MPO, controllable MPO, and our approaches. For MO-MPO and constrained MO-MPO, each dot corresponds to a separately-trained policy. For controllable MPO and controllable MO-MPO, five policies are evaluated by conditioning on a range of linearly-spaced preferences (which correspond to weights or KL-divergence bounds, respectively). The gray triangles denote the MO-MPO Pareto front.

by training policies with MO-MPO for a range of preference settings: $\epsilon_{\text{task}} = 0.1$ and $\epsilon_{\text{norm}}$ at linearly spaced intervals between $10^{-5}$ and $0.15$.

We trained preference-conditioned policies with five random initializations, for the same $\epsilon_{\text{task}}$ and $\epsilon_{\text{norm}}$ sampled uniformly from the same range; we call this approach controllable MO-MPO. This produces a Pareto front that is comparable to MO-MPO's, both in terms of coverage and solution quality (Fig. 5, middle right).

It is also possible to train preference-conditioned policies where the preferences correspond to weights for linear scalarization, rather than KL-divergence bounds as in MO-MPO. So as a baseline, we train weight-conditioned policies with MPO (Abdolmaleki et al., 2018); we call this baseline controllable MPO. These policies were conditioned on a $w_{\text{task}}$ of $1.0$ and $w_{\text{norm}}$ sampled uniformly between $0$ and $0.15$. Compared to controllable MO-MPO, we had to train controllable MPO for longer (900M actor steps instead of 500M) in order to reach reasonable task performance, and controllable MPO was still unable to find solutions with low action norm (Fig. 5, middle left). This is as expected: Abdolmaleki et al. (2020) show that MO-MPO outperforms MPO with linear scalarization on humanoid *run*, so one would expect controllable MO-MPO to also outperform controllable MPO. We tried running controllable MPO with higher weights, but these did not show any learning progress within 500M actor steps.

Policies trained with constrained MO-MPO for equality constraint thresholds from $-4$ to $-1$ also lie on the MO-MPO Pareto front (Fig. 5, right). We train a separate preference-conditioned action policy and preference policy for each unique constraint threshold. This supports that learning the preference distribution simultaneously while training the policy and Q-functions does not negatively impact the final solutions that are found.

## C.2 MPO LAGRANGIAN BASELINE

The baseline we use in our empirical evaluation is MPO with Lagrangian relaxation. We ran this baseline on all Safety Gym point mass tasks, using the default settings (including dense reward) so that

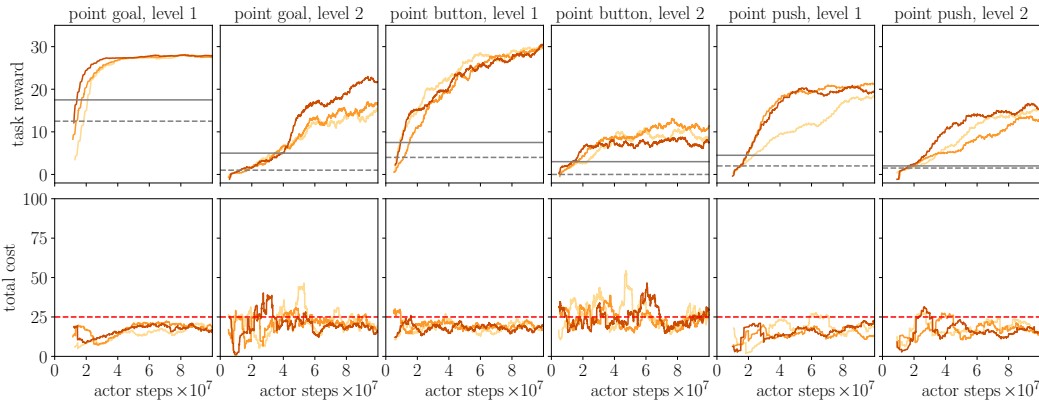

Figure 6: Top row: The task reward (averaged over episodes) achieved by policies trained with our MPO-Lagrangian baseline, over the course of training. The horizontal solid and dashed grey lines denote the final task reward obtained by the PPO-Lagrangian and TRPO-Lagrangian approaches, respectively, in Ray et al. (2019). Bottom row: The MPO-Lagrangian baseline meets the constraint of incurring less than 25 expected cumulative cost per episode, indicated by the dotted red line.

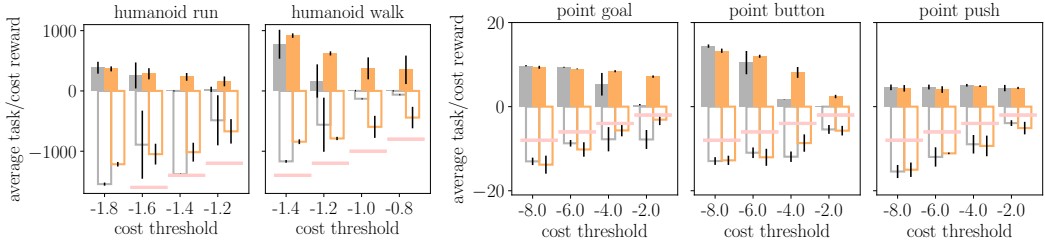

Figure 7: Constrained MO-MPO (in orange) achieves significantly higher task reward than the baseline (in gray) for the harder-to-satisfy constraint thresholds in humanoid **run** ($-1.4$ and $-1.2$), humanoid **walk** ($-1.2$, $-1.0$, and $-0.8$), point **goal** ($-4$ and $-2$), and point **button** ($-4$ and $-2$). For the other constraint thresholds, both approaches achieve similar task reward. In terms of satisfying constraint thresholds, our approach performs on-par with or better than the baseline. Each bar shows either the average task reward (solid bars) or cost (clear bars) per episode, averaged over five seeds. For both, higher is better. The error bars refer to standard deviation. The red lines denote the constraint thresholds: clear bars above the constraint threshold indicate that the constraint is met.

we can directly compare against the constraint-satisfying policies obtained in Ray et al. (2019).[11] Our MPO-Lagrangian baseline compares favorably with those in Ray et al. (2019), that instead combine Trust Region Policy Optimization (TRPO) and Proximal Policy Optimization (PPO) with Lagrangian relaxation (Fig. 6). This is with the caveat that policies were trained with TRPO-Lagrangian and PPO-Lagrangian for 10 million actor steps, whereas our training curves for MPO-Lagrangian go up to 100 million actor steps.

## C.3 MULTIPLE RANDOM SEEDS

To verify that the empirical results we observed in Sec. 5.1 are significant, we used our approach and the baseline to train policies for the same constraint threshold, starting from five random initializations. We evaluated this for four constraint thresholds per task, selected from the ranges in Table 3. Results are shown in Fig. 7.

Constrained MO-MPO achieves significantly higher task reward than the baseline for the harder-to-satisfy constraint thresholds in four out of the five tasks. For the other constraint thresholds, both

---

[11]Since we noticed that our Q-functions underestimate cost (Appendix C.4), we use a cost threshold of 15 for training, rather than the actual cost threshold of 25.

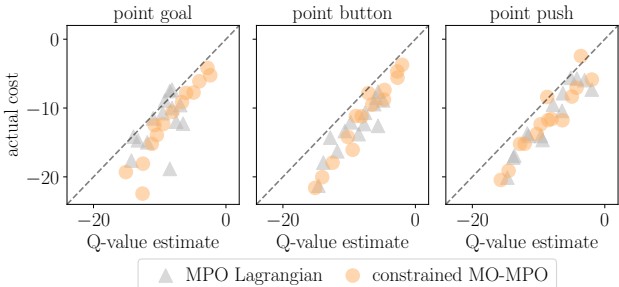

(a) The learned Q-function for cost underestimates the cost per episode. The actual cost is obtained by evaluating the deterministic policy after 200M actor steps, on the same 100 randomly-initialized environments.

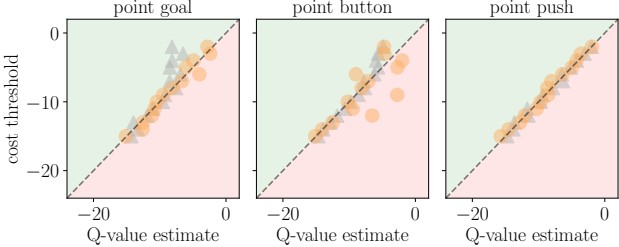

(b) For both approaches, almost all trained policies satisfy the constraint (i.e., are on or above the dotted line), according to the learned Q-values.

Figure 8: In the plots above, the Q-value estimate is obtained by averaging the Q-values for 100 batches of $(s, a)$ pairs from the replay buffer, after 200M actor steps of training. Each point corresponds to a separately-trained policy, for a different constraint threshold

approaches achieve similar task reward. In terms of satisfying constraint thresholds, our approach performs on-par with or better than the baseline.

As mentioned, the slight violation of constraint thresholds for the Safety Gym point tasks is due to underestimation of Q-values, rather than a limitation of the policy improvement method; this is described in the following subsection.

### C.4    Q-FUNCTION ESTIMATES

We noticed that for the Safety Gym point mass tasks, both our approach and the Lagrangian baseline train policies that slightly exceed the constraints (Fig. 1, bottom row). After digging deeper, we realized that this results from suboptimal Q-function learning, rather than suboptimal policy optimization. The learned Q-values for the cost consistently underestimate the actual cost incurred (Fig. 8a). The policy optimization finds a policy that meets the constraints, assuming that the learned Q-values for the cost are accurate (Fig. 8b).

Thus, better constraint satisfaction can be obtained by improving policy evaluation, so that estimated Q-values are more accurate—this is orthogonal to our proposed approach, which is focused on policy improvement.

### D    ALGORITHMIC DETAILS

In this section, we give details on how to learn the preference policy and provide an algorithm box for the policy improvement step. We also describe the overall procedure we use to collect data and learn from the data.

## D.1 LEARNING PREFERENCE-CONDITIONED POLICIES

As discussed in the main paper, we also learn a new preference policy $\pi_\psi(\boldsymbol{\epsilon})$ given the current preference policy $\pi_{\text{old}}(\boldsymbol{\epsilon})$ and fitness functions $f_k(\boldsymbol{\epsilon})$ (that evaluate $\boldsymbol{\epsilon}$ in terms of constraint satisfaction). One could add the fitness functions together to obtain $\sum_k f_k(\boldsymbol{\epsilon})$, and use any off-the-shelf RL algorithm to optimize for a new preference policy. On the other hand, this problem can be been as a multi-objective problem where the fitness for each constraint $k$ is an objective. To this end, we use MO-MPO for learning $\pi_\psi(\boldsymbol{\epsilon})$. Note that in this paper we have at most two constraints. In the case of a problem with one constraint (i.e., one "objective"), MO-MPO (Abdolmaleki et al., 2020) reduces to MPO (Abdolmaleki et al., 2018).

More formally, following the MO-MPO algorithm, we optimize the following constrained optimization problem, that can be solved via Lagrangian relaxation (Abdolmaleki et al., 2020):

$$\max_\psi \sum_{k=1}^{K} \text{KL}(p_k(\boldsymbol{\epsilon}) \| \pi_\psi(\boldsymbol{\epsilon})) \quad \text{s.t. } \text{KL}(\pi_{\text{old}}(\boldsymbol{\epsilon}) \| \pi_\psi(\boldsymbol{\epsilon})) < \delta \,,$$

where $\delta$ defines a trust region for more stable learning and $p_k(\boldsymbol{\epsilon})$ is a non-parametric improved policy for each objective, i.e,

$$p_k(\boldsymbol{\epsilon}) \propto \pi_{\text{old}}(\boldsymbol{\epsilon}) \exp\left(\frac{f_k(\boldsymbol{\epsilon})}{\varphi_k}\right) .$$

$\varphi_k$ is a temperature variable that we maintain for each objective (or fitness function) and is obtained by optimizing the convex dual function

$$g(\varphi_k) = \alpha_k + \log \int_{\boldsymbol{\epsilon}} \pi_{\text{old}}(\boldsymbol{\epsilon}) \exp\left(\frac{f_k(\boldsymbol{\epsilon})}{\varphi_k}\right) \text{d}\boldsymbol{\epsilon} \,, \tag{8}$$

where $\alpha_k$ for each objective $k$ defines a desired KL-divergence bound between the new improved policy $p_k(\boldsymbol{\epsilon})$ and current policy $\pi_{\text{old}}(\psi)$. We use the same value of $\alpha_k = 0.1$ for all fitness functions $k$ throughout the paper. For more details on MO-MPO, please refer to (Abdolmaleki et al., 2020).

Note that this section assumes that the preference policy is state-independent, i.e., $\pi(\boldsymbol{\epsilon})$. The optimization is analogous for a state-dependent preference policy $\pi(\boldsymbol{\epsilon}|s)$, which we consider in Sec. 5.3.

## D.2 GENERAL ALGORITHM

We maintain one online network and one target network for each Q-function, action policy and preference policy. We also maintain one online network for the temperature function. Target networks are updated every fixed number of steps by copying parameters from the online network. Online networks are updated using gradient descent in each learning iteration. We refer to the target networks by using the subscript/superscript "old" throughout the paper.

We use an asynchronous actor-learner setup. In this setup actors fetch policy parameters from the learner and act in the environment and write transitions to the replay buffer. Note that at the beginning of each episode, we first sample one preference parameter $\boldsymbol{\epsilon}$ from the preference policy, which remains fixed until the end of the episode. The learner uses the transitions in the replay buffer to update the Q-functions, policies and temperature functions. Algorithm 1 describes one step of policy improvement for constrained MO-MPO.

---

**Algorithm 1:** Constrained MO-MPO: One policy improvement step

---

1: **given** batch size ($L$), number of actions and $\boldsymbol{\epsilon}$ to sample ($M$), current policy $\pi_{\text{old}}(\boldsymbol{\epsilon})\pi_{\text{old}}(a|s,\boldsymbol{\epsilon})$, current ($K+1$) Q-functions $\{Q_k^{\text{old}}(s,a,\boldsymbol{\epsilon})\}_{k=0}^K$, temperature network $\eta_\omega(\boldsymbol{\epsilon})$, ($K$) constraint fitness functions $\{f_k(\boldsymbol{\epsilon})\}_{k=1}^K$, ($K$) temperature variables $\{\varphi_k\}_{k=1}^K$, ($K$) KL bounds $\{\alpha_k\}_{k=1}^K$, replay buffer $\mathcal{D}$, first-order gradient-based optimizer $\mathcal{O}$

2:

3: **initialize** $\pi_\theta(a|s,\boldsymbol{\epsilon})$ from the parameters of $\pi_{\text{old}}(a|s,\boldsymbol{\epsilon})$

4: **initialize** $\pi_\psi(\boldsymbol{\epsilon})$ from the parameters of $\pi_{\text{old}}(\boldsymbol{\epsilon})$

5: **repeat**

6:     **// Collect dataset** $\{s^i, \boldsymbol{\epsilon}^i, a^{ij}, Q_k^{ij}\}_{i,j,k}^{L,M,K}$**, where**

7:     **//** $L$ **states** $s^i \sim \mathcal{D}$

8:     **//** $L$ **preferences** $\boldsymbol{\epsilon}^i \sim \pi_{\text{old}}(\boldsymbol{\epsilon})$

9:     **//** $M$ **actions** $a^{ij} \sim \pi_{\text{old}}(a|s^i,\boldsymbol{\epsilon}^i)$ **and** $Q_k^{ij} = Q_k^{\text{old}}(s^i,\boldsymbol{\epsilon}^i,a^{ij})$

10:

11:     **// Compute (non-parametric) action distribution for each objective**

12:     $\delta_\omega \leftarrow \nabla_\omega \sum_k \frac{1}{L} \sum_i^L \eta_\omega(\boldsymbol{\epsilon}^i)[k]\left[\epsilon_k^i + \log \sum_j^M \frac{1}{M}\exp\left(\frac{Q_k^{ij}}{\eta_\omega(\boldsymbol{\epsilon}^i)[k]}\right)\right]$, where [k] is the index of the vector

13:     Update $\omega$ based on $\delta_\omega$, using optimizer $\mathcal{O}$

14:     **for** k = 0, ..., $K$ **do**

15:         $q_k^{ij} \propto \exp(\frac{Q_k^{ij}}{\eta_\omega(\boldsymbol{\epsilon}^i)[k]})$

16:     **end for**

17:

18:     **// Update action policy**

19:     $\delta_\theta \leftarrow -\nabla_\theta \sum_i^L \sum_j^M \sum_{k=0}^K q_k^{ij} \log \pi_\theta(a^{ij}|s^i,\boldsymbol{\epsilon}^i)$

20:     (subject to additional KL regularization)

21:     Update $\pi_\theta$ based on $\delta_\pi$, using optimizer $\mathcal{O}$

22:

23:

24:     **// Collect dataset** $\{\boldsymbol{\epsilon}^i, f_k^i\}_{i,k}^{M,K}$**, where**

25:     **//** $M$ **preferences** $\boldsymbol{\epsilon}^i \sim \pi_{\text{old}}(\boldsymbol{\epsilon})$ **and** $f_k^i = f_k(\boldsymbol{\epsilon}^i)$

26:

27:     **// Compute epsilon distribution for each constrained objective**

28:     **for** k = 1, ..., $K$ **do**

29:         $\delta_{\varphi_k} \leftarrow \nabla_{\varphi_k} \varphi_k \alpha_k + \varphi_k \log\left(\sum_i \frac{1}{M}\exp\left(\frac{f_k^i}{\varphi_k}\right)\right)$

30:         Update $\varphi_k$ based on $\delta_{\varphi_k}$, using optimizer $\mathcal{O}$

31:         $p_k^i \propto \exp(\frac{f_k^i}{\varphi_k})$

32:     **end for**

33:

34:     **// Update preference policy**

35:     $\delta_\psi \leftarrow -\nabla_\psi \sum_i^M \sum_{k=1}^K p_k^i \log \pi_\psi(\boldsymbol{\epsilon}^i)$

36:     (subject to additional KL regularization)

37:     Update $\pi_\psi$ based on $\delta_\psi$, using optimizer $\mathcal{O}$

38:

39: **until** fixed number of steps

40: return $\pi_{\text{old}}(a|s,\boldsymbol{\epsilon}) = \pi_\theta(a|s,\boldsymbol{\epsilon})$ and $\pi_{\text{old}}(\boldsymbol{\epsilon}) = \pi_\psi(\boldsymbol{\epsilon})$

---

