# OpenReview forum: "Explicit Pareto Front Optimization for Constrained Reinforcement Learning"
_ICLR.cc/2021/Conference — Reject_

### Official Review · AnonReviewer2 · 2020-10-22
**Contribution of the submission not clear enough**

**Rating:** 6
**Confidence:** 2

**Review:**

In Reinforcement Learning (RL) it is common that one searches for policies that optimize the reward and that meet certain constraints. A common approach for handling the constraints is Lagrangian relaxation, i.e., to incorporate the constraints in some way into the objective. In this submission, a different approach is suggested, in which the constraints are treated as additional objectives and then the Pareto front of policies is computed. This is reasonable as the optimal policy for a single objective that satisfies certain constraints must be Pareto optimal when the constraints are interpreted as additional objectives. It is then explained how the new approach can be implemented and several experiments are conducted on well-known data sets. It turns out that it is indeed beneficial for some data set to compute the set of Pareto-optimal policies instead of using Lagrangian relaxation because in some cases Lagrangian relaxation finds only the extreme policies while the Pareto front also contains intermediate policies that balance the criteria better.

I find the suggested approach very natural. The connection between constrained problems and the Pareto front is not very surprising and it is well-known and used extensively in the field of multi-objective optimization (for different problems). The writeup is rather condensed. I found it hard to follow Section 4 in detail (but I am not an expert in the field of RL). The experimental results look convincing. It is hard for me as as a non-expert in RL to give a clear recommendation for this submission. It is not clear to me if there is a novel contribution except for the idea to look at the Pareto front instead of a Lagrangian relaxation. It does not become clear to me if there is any non-trivial contribution in Section 4 or if the implementation follows more or less along the lines of previous RL implementations.

I got the impression (on page 3 this is said more or less explicitly) that the submission makes the assumption that for any Pareto-optimal solution there is a preference vector for which this particular solution is optimal. I don't think that this is true. This way one can characterize the convex hull of solutions but in general there can be Pareto-optimal solution that do not lie on the convex hull and for which there does not exist such a preference vector.

---

> ### Author Response · Authors · 2020-11-13
> **Author response to R2**
>
> Thank you for your review. We are glad that you find the approach natural, and found the empirical evaluation to be convincing. Please find below our response to your concerns.
>
> (1) Contribution not clear, compared to previous RL algorithms
> In Section 4, the extension from MO-MPO is non-trivial (Section 4.1), and the proposal to learn a preference policy (Section 4.2) is entirely novel. Please refer to our comment above to all reviewers, that summarizes our contributions and also details the significant differences between our approach versus MO-MPO.
>
> (2) Unsurprising connection between constrained problems and Pareto front
> We agree that the connection between multi-objective optimization and constrained problems is a clear and natural one to make. However, *in the context of RL*, the toolbox of existing constrained optimization methods is limited and the use of multi-objective approaches as an alternative is novel in this context. (If we have overlooked existing work, please point us to the relevant publications.)  To our knowledge, there is no pre-existing framework for applying multi-objective RL algorithms to constrained RL problems. A main contribution of this paper is to provide such a framework. We have updated the paper to clarify this.
>
> (3) Preferences for Pareto-optimal solutions
> If the preference vector encodes a linear weighting of objectives, then it is true that for Pareto-optimal solutions that are not on the convex hull, there is no preference vector for which the solution is optimal. However, we define the notion of "preference" more broadly, to indicate any encoding of trade-offs between objectives, not just linear scalarizations. For instance, if the preference vector corresponds to weights in Chebyshev scalarization, then the optimal solution for such a preference can indeed lie on concave portions of the Pareto front [1]. We have updated the paper to clarify this.
>
> [1] Moffaert et al. Scalarized multi-objective reinforcement learning: Novel design techniques. ADPRL 2013.

---

### Official Review · AnonReviewer4 · 2020-10-27
**Recommendation to Accept**

**Rating:** 7
**Confidence:** 3

**Review:**

The paper is interested in reinforcement learning where one needs to satisfy constraints (for instance energy spent) in addition to maximizing rewards. The proposed approach proposes to extend any method able to approximate the Pareto front of optimal policies by also learning portions of the front that satisfy user constraints (given that only small portion of the front may be valid). In particular, a previous approach MO-MPO is used to approximate the Pareto front in a first step and a second step learns another policy find preference vectors satisfying given constrains. Experiments are carried in two environments and show that the proposed method outperforms Lagrangian baselines (e.g. optimizing alternatively for reward and constrain violation) in particular in settings where constraints are harder to satisfy.

The main strength of this paper are its sound experimental results and overall improvements compared to the baselines studied. In addition, the paper is well-written and tackle a highly relevant problem. The weakness of the paper is that the method is a bit incremental compared to MO-MPO and that it may be hard to reproduce without code.

For these reasons, I recommend accepting the paper given its improvement over Lagrangian methods and its clarity. It would clearly help if the code allowing to rerun comparison could be provided so that future work can compare with the proposed approach.

# Additional feedback and suggestions

- The problem of concave Pareto front is indeed problematic for linear scalarization but recent work in scalarization may allow to bypass the issue [1]
- Figure 1: typo "contrianed"
- Figure 1: Is there a reason or an intuition on why outliers appear in MPO Lagrangian (bottom-left)? Is it because of the unstable behavior explained in Fig 2?
- "but as the constraint threshold becomes more difficult to satisfy, our approach dominates", except for "point push" this could be precised. Also is it because the front is closer to a convex?
- Figure 4: I suggest drawing a vertical line with the constraint at -3000, -2000 and -1500. Also it may be clearer to use a normalized action norm cost (e.g. -2.0 instead of -2000) to make it easier to compare with the cost constrain.

[1] Random Hypervolume Scalarizations for Provable Multi-Objective Black Box Optimization. Golovin et al. ICML2020

---

> ### Author Response · Authors · 2020-11-13
> **Author response to R4**
>
> Thank you for your review and suggestions. We are glad you agree the problem is highly relevant, found the paper well-written, and believe the empirical evaluation supports that our approach outperforms Lagrangian-based approaches. Please find below our response to your concerns and questions.
>
> (1) Incremental compared to MO-MPO
> Please refer to our comment above to all reviewers, that explains how controllable MO-MPO (i.e., our extended version of MO-MPO) is only one component of our more general framework, and can be seen as a design choice. It also enumerates the non-trivial differences between our extended version of MO-MPO versus MO-MPO.
>
> (2) Hard to reproduce without code
> We plan to release the code once the paper is published. Our implementation of constrained MO-MPO uses an open-source RL framework, and thus should be easy for others to use.
>
> (3) Recent work in scalarization
> We have updated the paper to cite [1] in the Related Work; thank you for bringing that to our attention. The advantage of our framework is that it is orthogonal to advances in multi-objective RL: if future multi-objective RL algorithms leverage [1] or other techniques to better find solutions on concave portions of the Pareto front, then our framework will enable applying these new algorithmic advances to solving constrained RL problems.
>
> (4) Outliers in Lagrangian baseline (Figure 1, bottom-left)
> Yes, these outliers are due to the unstable behavior of the Lagrangian baseline. The Lagrange multiplier must have a relatively small learning rate, to prevent it from changing too rapidly and thus destabilizing policy learning. The loss on the Lagrange multiplier means that when the constraint is met, the Lagrange multiplier decreases, and when the constraint is violated, the Lagrange multiplier increases. It takes some time for this to propagate into the policy, which results in the portions in Figure 2b (top row) where the average cost dips below (i.e., violates) the constraint threshold, indicated by the dotted line.
>
> (5) Where constrained MO-MPO dominates
> We will clarify that constrained MO-MPO and the baseline perform similarly on point push, including on lower-magnitude constraint thresholds. We hypothesize that this is because in the point push task, the cost does not conflict much with the task objective, so it is easier to find high-task-reward solutions that incur small cost. Whereas in a task like humanoid run/walk, trying to keep the action $\ell$2-norm (which is analogous to energy expenditure) below a threshold directly conflicts with running/walking, and thus it is harder to find high-task-reward solutions with small cost.
>
> (6) Clarity of Figure 4
> We have updated the Figure 4a plots to include vertical lines indicating the constraints. We previously considered using the average per-timestep action norm cost as the x-axis in the humanoid run/walk plots, but ultimately decided to stick with the convention of using average episode return, with an explanation in footnote 3.

---

> > ### Comment · AnonReviewer4 · 2020-11-23
> > **answer**
> >
> > Thank you for your answer and your clarification on the contribution. I encourage the author on following-up on delivering the code that will be valuable in my opinion.
> >
> > Regarding the discussion and the lack of responsiveness, I am really sorry to not have followed up earlier. In my case, I got all information I needed but I should have clearly stated so (I will consider updating my score and ponder your arguments).

---

> > > ### Author Response · Authors · 2020-11-23
> > > **Reply to R4**
> > >
> > > Thank you for your response! We are glad to hear that you have all the information you need, and we sincerely appreciate that you are considering an update to your score.
> > >
> > > Regarding open-sourcing, we completely agree that it is important to release code, and will deliver on our promise. The authors of MO-MPO recently open-sourced it on Github. We will open-source constrained MO-MPO by forking from that implementation, to make it easy for others to use and build on. (We would have loved to upload our code during this rebuttal period, but unfortunately it takes some time to prepare this.)

---

### Official Review · AnonReviewer1 · 2020-11-09
**Recommendation to Reject**

**Rating:** 4
**Confidence:** 3

**Review:**


Summary:

This paper introduced a general framework that incorporates multi-object reinforcement learning(MORL) perspective for constrained reinforcement learning to a policy set, called Pareto front, that meets the constrained. The author has instantiated a method based on the previous method MO-MPO. Compared to previous Lagrangian-based approaches, the proposed method has advantages in solution quality, stability, and sample-efficiency in few empirical environments.

##########################################################################
Reasons for score:

Overall, I'd vote for rejection for this paper. My major concerns lie in two aspects: 1) the innovation is minor compared to the previous method(MO-MPO). 2) the empirical evidence for the proposed method is marginal to me. Hopefully, the authors can address my concern in the rebuttal period.

Pros:
	1. The paper is straightforward and clear to read.

Cons:
	1. The motivation of the proposed framework is not strong. Besides that, the improvement of constrained MO-MPO is minor.
	2. The domains ( runs & walk) didn't illustrate the idea well. It is not clear to me what the constrained is. The advantage of constrained MO-MPO over lagrangian-based approaches is marginal. The advantages could vary from sed choice.
##########################################################################

---

> ### Author Response · Authors · 2020-11-13
> **Author response to R1**
>
> Thank you for your review. Please find below our response to your concerns and questions.
>
> (1) Minor innovation compared to MO-MPO
> Please refer to our comment above to all reviewers, that explains how controllable MO-MPO (i.e., our extended version of MO-MPO) is only one component of our more general framework, and can be seen as a design choice. It also enumerates the non-trivial differences between our extended version of MO-MPO versus MO-MPO.
>
> (2) Marginal empirical evidence
> When the constraint thresholds are easy to satisfy, constrained MO-MPO and the Lagrangian baseline indeed perform similarly. However, *when the constraint threshold is challenging to satisfy*, constrained MO-MPO outperforms the baseline in terms of solution quality, sample efficiency, and scaling up to multiple constraints. This is shown in several results figures:
> - Figure 1 (top row): For challenging (i.e., lower-magnitude) constraint thresholds, constrained MO-MPO policies achieve higher task reward than baseline policies.
> - Figure 2a: For constraint thresholds between $-1$ and $-2$ for humanoid run, constrained MO-MPO significantly outperforms the baseline, taking into account standard error bars.
> - Figure 3a: Regarding sample-efficiency, this improvement is even more stark mid-way through training, for humanoid run.
> - Figure 3c: In scaling up to multiple constraints, our approach outperforms the baseline for challenging constraint thresholds on one cost, and for *all* constraint thresholds on the other cost.
>
> We have updated the paper to clarify this and added Pareto plots (Figure 1, top row) that more clearly show the difference in performance for the harder-to-satisfy constraint thresholds.
>
> (3) Empirical advantage could come from seed choice
> Our experiments, as mentioned in the previous point, consistently show constrained MO-MPO has an empirical advantage over the Lagrangian baseline. Our approach outperforms the baseline in four out of five tasks from two open-source benchmark domains (DeepMind Control Suite and OpenAI's Safety Gym). These two domains, humanoid and point mass, have very different task rewards and costs. Thus it is unlikely that the consistent improved performance of our approach across multiple tasks is due to simply lucky seed choice, since we use the same seed (i.e., policy initialization) for all tasks.
>
> (4) Weak motivation for the proposed framework
> The vast majority of approaches for constrained RL rely on Lagrangian relaxation. As mentioned in the Introduction, Lagrangian-based approaches fall short in three main ways: 1) satisfying challenging constraint thresholds, 2) finding solutions in concave regions of the Pareto front, and 3) limited flexibility in terms of the choice of constraint. Applying multi-objective RL to solve constrained RL problems can handle these issues, and our empirical evaluation supports this.
>
> - Regarding the first issue, intuitively, the more difficult a constraint threshold is, the more it conflicts with the task objective. Lagrangian-based approaches will train policies that satisfy the constraint, but these approaches inhibit exploration (for the sake of meeting the constraint) and thus reach a suboptimal solution in terms of task performance. Our framework enables better exploration because it effectively maintains an ensemble over policies (for different preference settings), which reduces the chance of getting stuck in a local optimum. In addition, multi-objective RL specializes in dealing with objectives that conflict with each other, and since our framework leverages such algorithms, it is better equipped to find policies that perform well with respect to all objectives.
>
> - Regarding the second issue, in Lagrangian-based approaches, the relaxed objective is a weighted-sum of the task return and constraint violation, which assumes a convex Pareto front. Whereas there exist multi-objective RL approaches, such as MO-MPO, that can find solutions on concave regions of the Pareto front.
>
> - Regarding the third issue, our framework offers flexibility because we can accommodate different kinds of constraints by only changing the fitness function (for training the preference policy), without needing to modify the underlying algorithm. For example, this enables learning a *portion* of the Pareto front rather than a single point (as shown in Section 5.2 and Figure 4a).
>
> (5) Clarification of humanoid tasks
> In the humanoid run and walk tasks, the goal is to maximize task performance while meeting a constraint on the "energy" expended. This is a relevant setup because often in real-world robotics and control tasks, energy consumption needs to be considered. The cost for these humanoid tasks is the negative control norm ($|| a ||_2$). In this domain, the action is proportional to joint torques. The constraint is on the expected cost. For example, if the constraint is $-2$, that means the average control norm must be less than $2$. We have updated the paper to clarify this.

---

> > ### Author Response · Authors · 2020-11-23
> > **Additional experiments with multiple seeds**
> >
> > To address your concern regarding seed choice, we ran additional experiments for all five tasks (humanoid run/walk and point goal/button/push), where for each approach we train policies with five random seeds per constraint threshold. The results show that constrained MO-MPO has a clear empirical advantage: it significantly outperforms the Lagrangian baseline in four out of the five tasks for the harder-to-satisfy constraint thresholds, and performs on par for other constraint thresholds. We have updated the paper to include this experiment (please refer to Appendix C.3 and Figure 7). The plots include error bars that indicate standard deviation in task reward and cost across the five seeds.

---

### Author Response · Authors · 2020-11-13
**Author response, to all reviewers**

We thank the reviewers for their comments. We are happy that the reviewers found the paper well-written and clear, and agreed that our approach has advantages over typical Lagrangian-based approaches with respect to solution quality, stability, and sample-efficiency.

We were surprised that the reviewers' main concern is that our approach seems incremental when compared to MO-MPO, and would like to take this opportunity to address this concern.

*In fact, MO-MPO solves an entirely different problem (multi-objective RL), and thus cannot be directly applied to constrained RL. Our framework consists of several components; as just one of these components, we develop an extended version of MO-MPO and use it to learn a Pareto front curve.*

We argue that our approach is novel in the following ways:

1. To our knowledge, *no framework currently exists for applying multi-objective RL algorithms to constrained RL problems.* We present a general framework for this, by proposing to train a preference policy $\pi(\epsilon)$ that learns which preferences produce constraint-satisfying policies. This preference policy effectively decouples the constraint problem from the multi-objective problem. Our framework can thus leverage any multi-objective RL algorithm that can train a preference-conditioned action policy $\pi(a|s, \epsilon)$, to apply it to solve constrained RL problems.
2. MO-MPO only trains policies $\pi(a|s)$ for a single pre-specified preference setting $\epsilon$, so it cannot be directly used in our framework. To overcome this, our paper introduces a novel extension of MO-MPO, which we call controllable MO-MPO, to train preference-conditioned policies $\pi(a|s, \epsilon)$. Controllable MO-MPO trains a single policy that represents the entire Pareto front. This is a non-trivial extension to MO-MPO, that requires making the action policy and Q-functions preference-conditioned, replacing the scalar temperature with a preference-conditioned temperature network, and modifying the underlying MO-MPO optimization principles. We also introduce hindsight relabeling of preferences to stabilize off-policy learning and improve sample efficiency. In Appendix C.1, Figure 5 shows an empirical comparison between MO-MPO and our extended version, controllable MO-MPO.

In the revised paper, we have updated the Introduction and Approach sections to make these points more clear. We hope that by putting the reviewers' main concern to rest, we can change the outcome of this paper.

---

### Decision · Program_Chairs · 2021-01-07
**Final Decision**

**Decision:**

Reject

**Comment:**

Considering reviewers' comments and comparing with similar papers recently published or submitted, this is a good paper but hasn't reached the bar of ICLR.  We believe that the paper is not ready for publication yet, and strongly encourage the authors to use the reviewers' feedback to improve the work and resubmit to one of the upcoming conferences.